# Rapid stimulus-driven modulation of slow ocular position drifts

**Tatiana Malevich[1,2,3†], Antimo Buonocore[1,2†], Ziad M Hafed[1,2*]**

[1]Werner Reichardt Centre for Integrative Neuroscience, Tuebingen University, Tuebingen, Germany; [2]Hertie Institute for Clinical Brain Research, Tuebingen University, Tuebingen, Germany; [3]Graduate School of Neural and Behavioural Sciences, International Max-Planck Research School, Tuebingen University, Tuebingen, Germany

**Abstract** The eyes are never still during maintained gaze fixation. When microsaccades are not occurring, ocular position exhibits continuous slow changes, often referred to as drifts. Unlike microsaccades, drifts remain to be viewed as largely random eye movements. Here we found that ocular position drifts can, instead, be very systematically stimulus-driven, and with very short latencies. We used highly precise eye tracking in three well trained macaque monkeys and found that even fleeting (~8 ms duration) stimulus presentations can robustly trigger transient and stimulus-specific modulations of ocular position drifts, and with only approximately 60 ms latency. Such drift responses are binocular, and they are most effectively elicited with large stimuli of low spatial frequency. Intriguingly, the drift responses exhibit some image pattern selectivity, and they are not explained by convergence responses, pupil constrictions, head movements, or starting eye positions. Ocular position drifts have very rapid access to exogenous visual information.

**\*For correspondence:**
ziad.m.hafed@cin.uni-tuebingen.
de

†These authors contributed
equally to this work

**Competing interests:** The
authors declare that no
competing interests exist.

**Reviewing editor:** Emilio
Salinas, Wake Forest School of
Medicine, United States

## Introduction

Eye position changes subtly even when perfect gaze fixation is attempted (*Barlow, 1952*; *Nachmias, 1959*; *Nachmias, 1961*; *Ratliff and Riggs, 1950*; *Steinman et al., 1973*). Such 'fixational' eye movement comes in two primary flavors: microsaccades, which resemble large saccades (*Hafed, 2011*; *Rolfs, 2009*; *Zuber et al., 1965*) and rapidly shift gaze position by a minute amount; and ocular position drifts, which are even smaller and slower eye movements (*Barlow, 1952*; *Ditchburn and Ginsborg, 1953*; *Martins et al., 1985*; *Nachmias, 1959*; *Nachmias, 1961*; *Poletti and Rucci, 2016*; *Ratliff and Riggs, 1950*; *Skinner et al., 2019*; *St Cyr and Fender, 1969*).

The mechanisms for generating and influencing microsaccades have received much attention (*Hafed, 2011*; *Hafed et al., 2015*; *Krauzlis et al., 2017*). Despite contrary ideas in the past century, a now accepted property of microsaccades is that, like larger saccades, they are not random but are very systematically (*Engbert, 2006*; *Engbert and Kliegl, 2003*; *Hafed and Clark, 2002*; *Hafed and Ignashchenkova, 2013*; *Hafed et al., 2011*; *Ko et al., 2010*; *Rolfs et al., 2008*; *Thaler et al., 2013*; *Tian et al., 2016*) and rapidly (*Buonocore et al., 2017a*; *Hafed and Ignashchenkova, 2013*; *Rolfs et al., 2008*; *Tian et al., 2016*; *Tian et al., 2018*) influenced by peripheral as well as foveal visual stimuli, among other factors (*Hafed et al., 2015*; *Willeke et al., 2019*). In stark contrast, the brain mechanisms for controlling ocular position drifts are unknown; these movements continue to be thought of as random, often being modeled as random walk processes (*Burak et al., 2010*; *Engbert and Kliegl, 2004*; *Engbert et al., 2011*; *Herrmann et al., 2017*; *Kuang et al., 2012*). This is despite the fact that slow ocular position drifts exhibit behavioral characteristics (e.g. *Ditchburn and Ginsborg, 1953*; *Nachmias, 1961*) supporting some level of central nervous system control over them. For example, these eye movements aid in stabilizing eye position near a certain location (e.g. *Epelboim and Kowler, 1993*); they can compensate for subtle head movements

**eLife digest** Vision is a highly complex, active process. As we observe and interact with the world around us, we constantly use eye movements to capture the visual information we need. In fact, our eyes continue to make tiny, unconscious movements even when we try to fix our gaze on something.

There are two main types of tiny eye movements. The first kind, so called microsaccades, are fast, microscopic flicks that happen every second or half-second. The other kind, termed drift, is a slower, gradual motion that takes place between microsaccades, or at any time when other eye movements are not happening. However, we know far less about drifts than about any other eye movements: both the reason why they occur and the brain mechanisms controlling them are still unclear.

Many scientists think that drifts are largely random movements, without any set direction. However, eye drifts do sometimes align with other behaviours – for example, they can help compensate for small, subtle head movements – suggesting that drifts may not be completely random after all. Malevich, Buonocore and Hafed therefore set out to test the hypothesis that eye drifts could, under the right circumstances, in fact be highly directed movements.

These experiments used precise sensors to track eye movements in macaque monkeys, which had been trained to fix their gaze on images or shapes (stimuli) presented on a screen. This revealed that presenting new stimuli, even for a few thousandths of a second, could repeatedly trigger drifts. This reaction also happened quickly, starting less than one hundredth of a second after presentation of the stimulus.

Further tests, using different images, revealed that the drifts were not only simply reacting to any new stimuli but also appeared to be a partially selective response to specific types of images. These tended to have larger features and less fine detail. For example, a picture of a landscape with large swaths of sky or hilltops would much more reliably trigger the eye drifts than a finely detailed checkerboard pattern, with many small squares alternating between black and white. These results suggested that drifts, far from being random movements, could be another tool for the brain to process visual information.

This work sheds new light on the potential role of eye movements in vision, and adds another layer of complexity to the question of how we see. Malevich et al. hope that this study will inspire further research into the brain mechanisms behind ocular drifts.

through mechanisms likely related to the vestibular ocular reflex (e.g. *Ditchburn and Ginsborg, 1953*; *Poletti et al., 2015*; *Schor and Westall, 1984*; *Skavenski et al., 1979*; *Steinman and Collewijn, 1980*); and they also appear to be associated with internal knowledge of their occurrence for regulating the perception (or lack thereof) of the image motions that they cause on the retina (e.g. *Poletti et al., 2010*).

All of the above hallmarks of brain control over drifts, however, still do not rule out the possibility that such control may be somewhat 'loose'. For example, the movements could still be random walks but nonetheless have statistics sufficient to modulate retinal image information for the purposes of aiding spatial vision (*Kuang et al., 2012*; *Rucci and Victor, 2015*). This leaves the question of how precisely these eye movements may potentially be controlled still unanswered. Here, we used precise eye tracking in three well trained rhesus macaque monkeys to demonstrate that ocular position drifts can exhibit highly systematic stimulus-driven modulations in both speed and direction. These modulations have a very short latency, and they are stimulus-tuned and binocular. These modulations are also independent of head movements, convergence responses, and starting eye positions. Moreover, their amplitudes are sufficient to move images across individual cone photoreceptors in the fovea or multiple rod photoreceptors in some peripheral zones.

Our results, coupled with evidence that drift statistics adapt to a variety of behavioral task constraints (*Chen and Hafed, 2013*; *Cherici et al., 2012*; *Epelboim and Kowler, 1993*; *Intoy and Rucci, 2020*; *Poletti et al., 2015*; *Poletti et al., 2010*; *Shelchkova et al., 2019*; *Skinner et al., 2019*; *Tian et al., 2018*), strongly motivate deeper research into the neurophysiological mechanisms controlling incessant ocular position drifts in between saccades.

## Results

### Short-latency, stimulus-driven ocular position drift response

We analyzed microsaccade-free fixation after a visual transient in monkeys highly trained to fixate on tiny targets. In a first experiment, each monkey fixated a white spot over a gray background spanning approximately + / - 15 deg horizontally and + / - 11 deg vertically. At a random time, the display changed: one entire half (right or left) became black (split view stimulus); and a half-circle of 0.74 deg radius around the fixated position remained gray, in order to maintain view of the stable fixation spot (Methods). We explicitly used such a patterned stimulus configuration (as opposed to just a flash as in some subsequent experiments described below) in order to explore any potential stimulus selectivity of ocular position drifts, as we elaborate on shortly. In trials without microsaccades (−100 to 200 ms from stimulus onset), a highly repeatable short-latency drift response occurred. This drift response is illustrated in *Figure 1A,B* for monkey A, in which we tracked the left eye using the precise scleral search coil technique (*Fuchs and Robinson, 1966*; *Judge et al., 1980*); the response measured in the two other monkeys, in which we measured right eye position, is shown in *Figure 1—figure supplement 1*. Regardless of the tracked eye, and regardless of the steady-state stereotypical drift direction exhibited individually by a given monkey, there was always a robust upward position drift after stimulus onset. Importantly, it was clearly visible at the individual trial

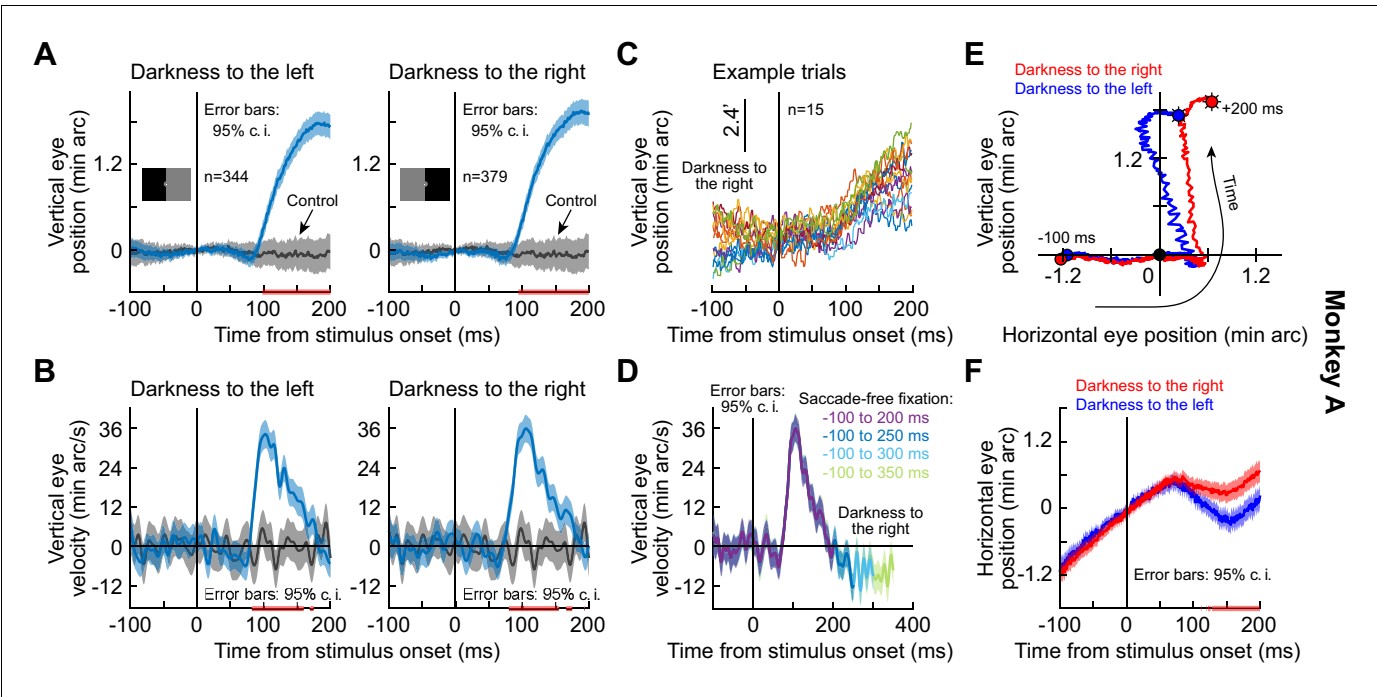

**Figure 1.** Short-latency, stimulus-driven ocular position drift response. (A) Vertical eye position (plus 95% confidence intervals) after split view stimulus onset. Compared to the no-stimulus condition (gray), the eye drifted upwards with short latency after stimulus onset. This happened for both right and left darkness stimuli (the two panels). We vertically aligned all starting eye positions at time 0 before averaging trials, to highlight the systematic change in eye position shortly after stimulus onset. (B) Same data as in (A) but for vertical eye velocity. A clear velocity pulse (much slower than that of microsaccades/saccades) is visible. (C) Individual trials demonstrating the drift response. The vertical position of each curve was jittered across the shown trials to facilitate visibility. (D) When we included progressively longer intervals of saccade-free fixation, we confirmed that the drift response (e.g. B) is transient. (E, F) Relative to starting eye position at stimulus onset, the drift response was predominantly vertical, but the horizontal component of eye position was also stimulus-dependent. Error bars in (F) denote 95% confidence intervals. The trajectories in (E) are two-dimensional plots of the average curves in (A), (F). All pink lines on x-axes indicate intervals in which the 95% confidence intervals of the two compared curves did not overlap. Also see *Figure 1—figure supplements 1* and *2*.

The online version of this article includes the following source data and figure supplement(s) for figure 1:

**Source data 1.** Excel table with the source data for this figure.
**Figure supplement 1.** Short-latency, stimulus-driven ocular position drift response in two additional monkeys besides the monkey shown in *Figure 1*.
**Figure supplement 2.** The ocular position drift response occurred earlier than pupil diameter constriction occurring after stimulus onset.

level (*Figure 1C*, *Figure 1—figure supplement 1C,I*). In one monkey (N), this upward drift response was sometimes initiated first by a much more short-lived and downward shift of eye position (of very small magnitude) before the flip to the upward drift (*Figure 1—figure supplement 1G–L*).

We statistically assessed the onset and duration of the drift response in each monkey by computing 95% confidence intervals (Methods) around the average eye velocity traces. When the 95% confidence intervals between the stimulus and control velocity traces did not overlap for at least 20 consecutive milliseconds, we deemed the velocity response to be significant. In monkey A, the upward velocity pulse started at ~80 ms and lasted for ~60 ms (*Figure 1B*). These values were ~60 ms and ~90 ms, respectively, for monkey M (*Figure 1—figure supplement 1B*) and ~70 ms and ~100 ms, respectively, for monkey N (*Figure 1—figure supplement 1H*). Note that *Figure 1A* and *Figure 1—figure supplement 1A,G* also show significance intervals for the eye position traces for completeness, although these provide a more conservative estimate of response onset time (Methods).

In all three monkeys, the drift response consisted of a predominantly upward velocity pulse reaching a peak speed of ~33–45 min arc/s, or ~0.55–0.75 deg/s (*Figure 1B,D*, *Figure 1—figure supplement 1B,D,H,J*) and a resultant displacement of ~2–3 min arc, or 0.034–0.05 deg (*Figure 1A*, *Figure 1—figure supplement 1A,G*). Such speeds and displacements are well within the detectability ranges of the visual system. For example, foveal cone photoreceptor separation in the rhesus macaque retina is ~1 min arc (*Rolls and Cowey, 1970*), and rod photoreceptor density can be even higher in the rod peak region (*Wikler et al., 1990*). Thus, the stimulus-driven drift response clearly modulates input images to the visual system.

The drift response was also transient: when we considered longer periods of saccade-free fixation, we did not notice increasing durations of drift response. Instead, the eye velocity 'pulse' subsided, and eye position reverted back to its steady-state drift direction that was prominent, and stereotyped, in each monkey (*Figure 1D*, *Figure 1—figure supplement 1D,J*).

The drift response also showed horizontal direction dependence on the stimulus properties, and this is why our use of the split view stimulus in the first experiment was relevant. In all three monkeys, the horizontal component of the eye movement during the drift pulse was systematically biased by the side of darkness (*Figure 1E,F*, *Figure 1—figure supplement 1E,F,K,L*): the upward drift tilted rightward when the darkness was on the right of gaze and leftward when it was on the left, and these results were statistically significant (pink intervals on all relevant x-axes of plots; Methods). The horizontal component of eye position therefore reacted to the spatial position of the luminance transient. This was our first evidence that the drift response can be stimulus-specific, a property that we revisit below.

## The drift response is neither an eye tracking artifact nor a convergence reflex

We considered the possibility that the stimulus-driven drift response was part of a near reflex triad (convergence, accommodation, and pupil constriction) (*Myers and Stark, 1990*; *Nachmias, 1961*). For example, the luminance transient may have been perceived by the monkey as a change in depth plane. However, this is unlikely to explain our observations: the horizontal bias as a function of stimulus appearance (e.g. *Figure 1E,F*) occurred regardless of whether we tracked the right (monkeys M and N) or left (monkey A) eye. For example, if the drift was part of a convergence movement, it would not be biased rightward when tracked in the right eye (monkey M). We also explicitly tested for convergence in a later experiment, described below, in which we measured eye movements binocularly.

Our results also cannot be explained by variations in pupil diameter, which can cause artifactual eye drift measurements in video-based eye trackers (*Choe et al., 2016*; *Drewes et al., 2014*; *Hooge et al., 2019*; *Kimmel et al., 2012*; *Wildenmann and Schaeffel, 2013*; *Wyatt, 2010*); in all of our analyses, we measured eye position using scleral search coils. Moreover, pupillary responses to stimulus transients exhibit significantly slower dynamics (*Clarke et al., 2003*; *Pong and Fuchs, 2000*). To confirm this, we performed a control experiment in which we explicitly measured pupil diameter in one monkey (A) using a video-based eye tracker. Using a stimulus manipulation in which the upward drift response was clearly visible in the (left) eye (tracked with a scleral search coil), simultaneous measurement of pupil diameter in the other eye (tracked with a video-based eye tracker; Methods) revealed that the pupillary constriction caused by stimulus onset occurred later than the

ocular position drift response (*Figure 1—figure supplement 2*). It was futile to look for the drift response in the right eye, confirming that video-based eye trackers are not capable of resolving such small detail (*Choe et al., 2016*; *Kimmel et al., 2012*; *Wyatt, 2010*).

## The drift response is a binocular eye movement

To directly investigate the convergence question, we next implanted one monkey (M) with a second scleral search coil, this time in the left eye. We could therefore measure binocular movements simultaneously. The monkey performed a new task in which we avoided a prolonged period of darkness in half of the display. Instead, the monkey steadily fixated with the original gray background. Then, for only one display frame (~8 ms), we flipped the luminance of the gray screen to black before it returned to gray again (Methods). We still observed a similar drift response (*Figure 2A,C*). Importantly, the drift response occurred in both eyes simultaneously (*Figure 2A,C*), and the horizontal component of eye position did not show an appreciable convergence response of similar magnitude (*Figure 2B,D*). We confirmed this statistically; in a time interval capturing the drift response (50–140 ms), there was no statistically significant difference between right and left peak vertical or peak horizontal eye velocity (p=0.249, Z = 1.15 for vertical and p=0.8321, Z = 0.21 for horizontal; Wilcoxon

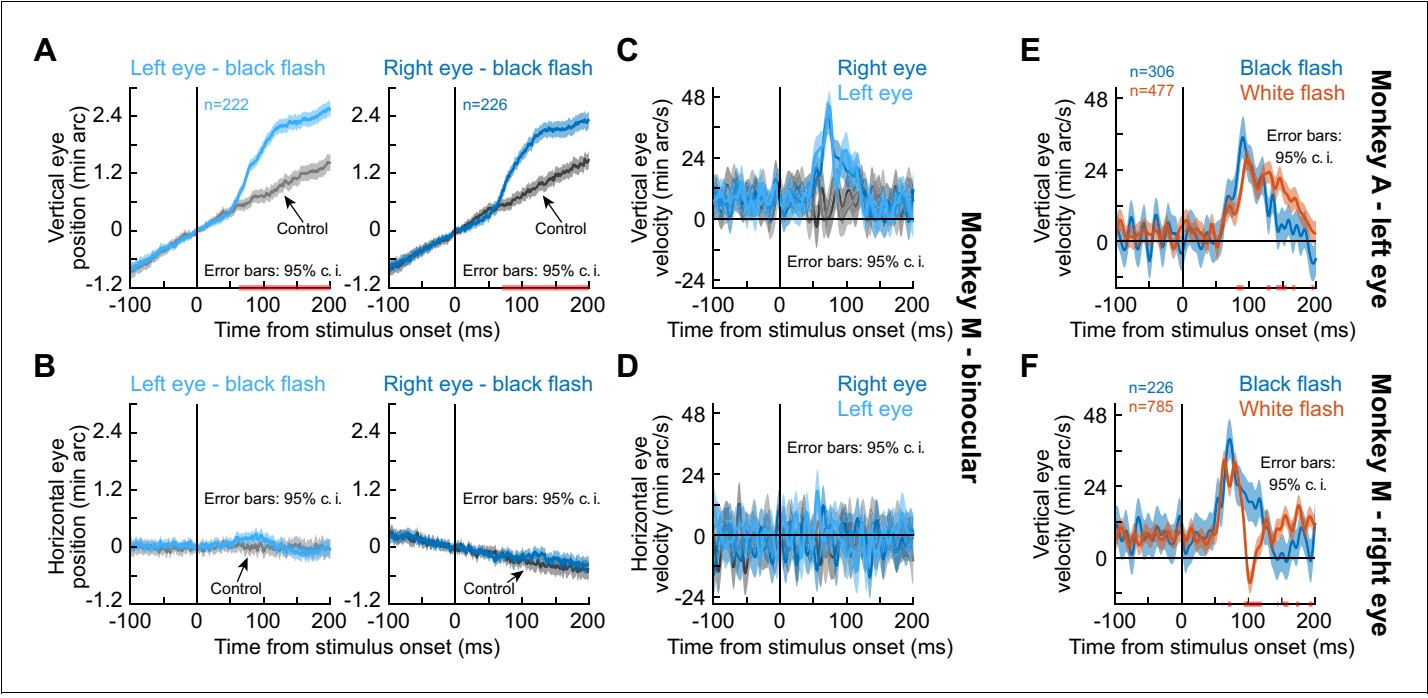

**Figure 2.** The drift response is a binocular eye movement. (A) Vertical eye position from monkey M during simultaneous measurements of the right and left eyes. The upward drift response (for a brief ~8 ms black flash covering the whole display) occurred binocularly. Note that this response superseded the monkey's stereotypical baseline slow upward drift, which was much slower (each monkey had a stereotypical baseline drift direction; *Figure 1*, *Figure 1—figure supplement 1*). (B) Horizontal eye position showed no evidence of convergence. (C, D) Same data with vertical and horizontal eye velocity, respectively. The drift response (C) was not associated with a convergence response (D). The lack of horizontal pink lines on the x-axes of (C), (D) confirm that both eyes moved similarly. (E) In monkey A, we confirmed that the same paradigm (brief black flash) was still sufficient (bluish curve) to cause an upward drift response. When we switched the flash to white (reddish curve), the drift response still occurred. (F) Monkey M also showed a drift response for a white flash. Error bars in all panels denote 95% confidence intervals, and pink lines on the x-axes show periods in which the 95% confidence intervals of the two compared curves did not overlap.

The online version of this article includes the following source data and figure supplement(s) for figure 2:

**Source data 1.** Excel table with the source data for this figure.
**Figure supplement 1.** The drift response is independent of head movements.
**Figure supplement 2.** Small localized stimulus onsets had minimal effects on ocular position drifts when compared to larger flashes.
**Figure supplement 3.** Independence of the ocular position drift response from starting eye position.
**Figure supplement 4.** The drift response was not systematically related to the direction of the first microsaccade to occur after it.

rank sum test). Therefore, the stimulus-driven drift response (*Figures 1* and *2*) is a binocular eye movement phenomenon that is independent of a convergence response.

Naturally, we also tested a second monkey (A) on this new black fixation flash paradigm (albeit monocularly), and we still observed the drift response (*Figure 2E*, bluish curve). Therefore, the drift response is not a peculiarity of the split view stimulus, and it still occurred in two monkeys under a completely different protocol.

## The drift response is not the same as a steady-state up-shift in gaze in darkness

We next asked whether our observations were related to a previously reported steady-state upward eye position deviation in darkness (*Barash et al., 1998*; *Goffart et al., 2006*; *Snodderly, 1987*). After all, our split view stimulus introduced substantial darkness (in an already dark laboratory), and the fixation flash paradigm (*Figure 2A–D*) also introduced darkness, although for just a fleeting moment. To test this, we ran the same two monkeys (A and M) on a third condition in which the entire display went from gray to white instead of black (again, for only ~8 ms; Methods). For both white and black flashes, a similar stimulus-driven transient ocular position drift response occurred (*Figure 2E,F*). Statistically, peak eye velocity was significantly higher with black than white flashes in both monkeys ($p=4.2\times10^{-14}$, $Z = 7.55$ for monkey A and $p=4.3\times10^{-5}$, $Z = 4.09$ for monkey M; Wilcoxon rank sum test), suggesting a secondary impact of stimulus polarity on the oculomotor system (*Malevich et al., 2020*).

## The drift response is not associated with a reflexive head movement

Our results so far involved measuring only eye position; our monkeys' heads were robustly fixed with a metal holder (Methods). Nonetheless, it could be argued that the drift response is potentially a stabilizing reflex due to stimulus-driven head movements (e.g. due to, say, startle by the monkeys after sudden flashes). Indeed, microscopic head movements (*Corneil and Munoz, 2014*) may be associated with small compensatory eye movements (*Ditchburn and Ginsborg, 1953*; *Poletti et al., 2015*; *Schor and Westall, 1984*; *Skavenski et al., 1979*; *Steinman and Collewijn, 1980*). Alternatively, if the head were to move in a stimulus-driven manner, then the drift response could be an artifactual measurement aberration due to movement of the head within the fixed magnetic fields of the scleral search coil system used to track eye movements; such potential head movement would, then, alter the measured eye position, but only due to physical displacement of the eye (by the head movement) in the magnetic fields and not because of a genuine eyeball rotation.

While it would be surprising to get a startle-induced head movement with such short latencies for our stimuli (some of which were very subtle and brief; *Figure 2*), we nonetheless performed additional control experiments in which we simultaneously measured eye and head position. We placed a 'head coil' in three different positions/orientations on the monkey's head, across three different sets of experiments, and we digitized the head coil output (into two channels) like we digitized the eye coil measurements into two channels (Methods). We used three different head coil positions on the head (in three sets of experiments) because head-fixation apparati and recording chambers implanted on the head resulted in non-standard head coil orientations relative to the magnetic fields around the head used to induce coil currents; we therefore wanted to ensure that there was no particular axis of head movement that we would potentially miss from using only one potential unfortunate head coil orientation in one given experiment. We also used the same black fixation flash paradigm of *Figure 2*, and we performed these additional control experiments in monkey M.

For all three sets of new experiments, we replicated the drift response observation in the same monkey (*Figure 2—figure supplement 1A*). Simultaneous measurements of head position (*Figure 2—figure supplement 1B,C*) did not reveal any concomitant head movements. To assess this statistically, we used a velocity estimate for both eye and head position measurements in each experiment (similar to our earlier analyses above). We specifically defined a pre-stimulus baseline interval and a post-stimulus 'drift' interval (indicated in pink in *Figure 2—figure supplement 1*). Within each interval, we averaged the velocity measure, resulting in a paired measurement between intervals across trials. For eye movements, there was a significant difference between vertical eye velocity during the post-stimulus drift interval and the pre-stimulus baseline interval ($p=5.55\times10^{-19}$ and $Z = -8.9$, $p=3.27\times10^{-18}$ and $Z = -8.7$, $p=1.05\times10^{-13}$ and $Z = -7.44$, respectively, for the three

head coil positions in *Figure 2—figure supplement 1A*; Wilcoxon rank sum test). This directly replicates our earlier drift response observations in multiple monkeys above. For head movements, none of the head coil positions that we tested resulted in substantial movements after stimulus onset (p=0.1394 to 0.8911 and Z = −0.91 to 1.48 across all positions and channels in *Figure 2—figure supplement 1B,C*; Wilcoxon rank sum test). These results mean that the head did not move at the same time as the ocular position drift response, ruling out both a compensatory eye movement explanation as well as an artifactual change in sensed position due to displacement of the eye's 3-dimensional position in the search coil system's magnetic fields.

## The drift response is feature-tuned

A significant swath (i.e. low spatial frequency) of luminance transience seemed necessary for the drift response. We did not observe the short-latency response as strongly with small localized targets briefly flashed to the right or left of fixation (*Figure 2—figure supplement 2*). Our earlier experiments with small localized stimuli (*Tian et al., 2018*) also only revealed a later position modulation than the drift response being characterized in the current study (which could have additionally been caused by stimulus-modulated microsaccades).

Given that visual responses in oculomotor areas like the superior colliculus (SC) preferentially represent low spatial frequencies (*Chen and Hafed, 2018*), we therefore wondered whether the ocular position drift response could exhibit tuning to spatial frequency. We replicated the same full-screen flash paradigm in one monkey (M) but now with different spatial frequencies (0.55–6.8 cycles/deg; cpd). We saw clear preference for low spatial frequencies in the drift response, both in response latency and magnitude (*Figure 3A*). To confirm this statistically, we searched for peak eye velocity in the interval 80–140 ms after stimulus onset (*Figure 3*). We then characterized peak eye velocity and peak eye velocity latency as a function of spatial frequency using a Kruskal-Wallis test. There was an effect of spatial frequency on both peak velocity (p=2.2×10⁻¹⁶, $\chi^2(4)$=79.49) and peak velocity latency (p=3.1×10⁻¹⁷, $\chi^2(4)$=83.56). Post-hoc tests revealed that the drift response was earliest (adjusted p<9.3×10⁻⁸ against all other spatial frequencies) and strongest (adjusted p<0.0021 against the third to fifth spatial frequencies) for the lowest spatial frequency. With our split view stimulus and display geometry (tested in all three monkeys), the effective spatial frequency was even lower than 0.55 cpd, and the drift response appeared to be

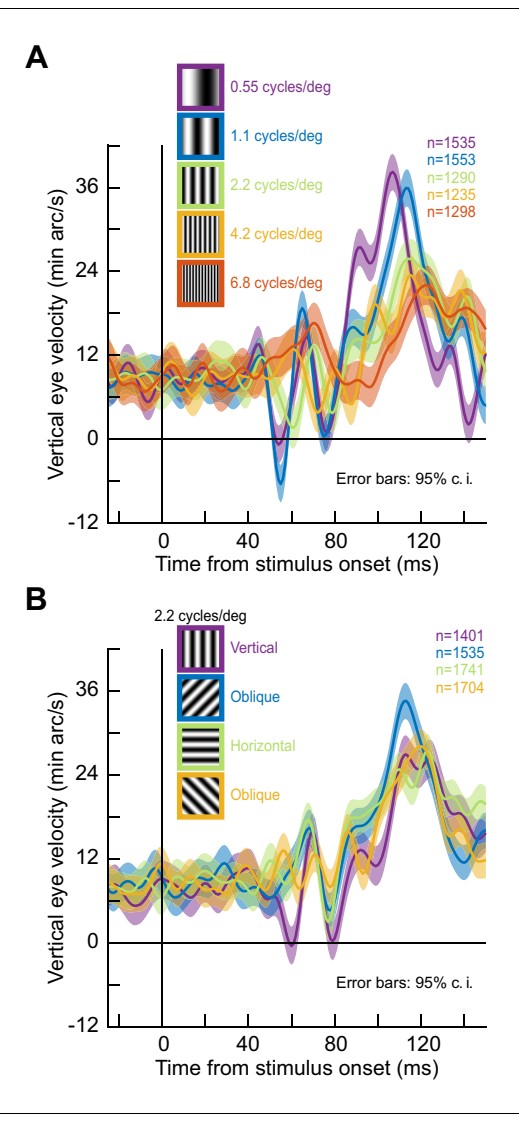

**Figure 3.** Stimulus tuning of the ocular position drift response. (**A**) We replaced the full-screen flash of *Figure 2* with a vertical grating of different spatial frequencies (Methods). The upward velocity pulse depended on spatial frequency; it occurred earlier, and had stronger amplitude, for low spatial frequencies. This is reminiscent of visual neural activity in the superior colliculus, an important oculomotor control circuit (*Chen et al., 2018*). (**B**) Similarly, when we fixed spatial frequency and altered grating orientation, the drift response was also modulated. Ocular position drifts are therefore sensitive to stimulus manipulations. Error bars denote 95% confidence intervals.

The online version of this article includes the following source data for figure 3:

**Source data 1.** Excel table with the source data for this figure.

the strongest (*Figure 1*, *Figure 1—figure supplement 1*). Therefore, the ocular position drift response not only occurs early enough to reflect potential effects of early sensory responses in the oculomotor system (e.g. SC visual bursts; *Chen et al., 2018*), but it is also sensitive to stimulus specifics. In fact, when we ran yet another experiment with grating orientation, rather than spatial frequency, the drift response of the same monkey was again altered, and it displayed a form of orientation tuning (*Figure 3B*; p=0.01, $\chi^2(4)$=11.0 for peak velocity and p=0.0004, $\chi^2(4)$=18.5 for peak velocity latency; Kruskal-Wallis test). Thus, ocular position drifts are sensitive, with short latency, to stimulus transients and their image features.

### The drift response happens at a time of complete saccade inhibition and is independent of starting eye position

In all of our analyses above, we isolated the properties of the drift response by analyzing saccade-free epochs. However, in reality, saccades could still occur even in our fixation paradigms. We therefore now considered the trials in which fixational microsaccades did occur. We found a remarkably complementary nature between the timing of microsaccades and the timing of the ocular position drift response. Specifically, it was long known that saccade and microsaccade likelihood decreases dramatically shortly after visual transients (*Buonocore et al., 2017a*; *Buonocore and McIntosh, 2008*; *Edelman and Xu, 2009*; *Engbert and Kliegl, 2003*; *Hafed and Ignashchenkova, 2013*; *Hafed et al., 2011*; *Reingold and Stampe, 2002*). We replicated this finding in all of our three monkeys, and we also compared the timing of the drift response to the timing of such saccadic inhibition (*Figure 4A–F*; split view stimulus). The drift response occurred when it was least likely to observe microsaccades. Even though the mechanisms of (micro)saccadic inhibition are still debated, we recently hypothesized that sensory transients in the oculomotor system (e.g. SC) play a particularly important role (*Buonocore et al., 2017a*). The stimulus dependence of the ocular drift response (e.g. *Figure 3*) and its timing relative to saccadic inhibition (and indeed relative to the timing of early sensory responses in the oculomotor system; *Chen et al., 2018*) suggest that the ocular drift response is mechanistically linked to the circuits mediating saccadic inhibition; and particularly to the short-latency visual responses in these circuits. That is, stimulus onset triggers both saccadic inhibition and drift response. Future neurophysiological experiments will have to directly test this hypothesis.

Moreover, we found that the drift response could still occur with nearby microsaccades before or after it. Specifically, for a given experiment, we picked trials in which microsaccades occurred within a given time range (e.g. up to 0 ms relative to stimulus onset). We then plotted the average eye velocity in the remainder of these trials (i.e. the portion of eye velocity that was non-saccadic). For different microsaccade times relative to stimulus onset, we observed similar amplitudes of the ocular drift response (*Figure 4G–I*). This also happened in our other tasks (e.g. *Figure 4—figure supplement 1*). Interestingly, the amplitude of the drift response seemed similar regardless of the timing of nearby microsaccades (*Figure 4G–I*). This is an interesting contrast to our earlier findings that amplitudes of the faster (and later) initial ocular following responses after full-screen motion pulses are strongly modulated when microsaccades happen near the time of stimulus (i.e. full-screen motion) onsets (*Chen and Hafed, 2013*).

Finally, we checked whether the drift response could still occur with different starting eye positions. We divided trials based on eye position being above or below median vertical eye position across trials (Methods), and the same upward drift response was present (*Figure 2—figure supplement 3* shows these results from the black fixation flash paradigm in monkeys A and M). And, the drift response was too small to even elicit corrective downward microsaccades after it occurred: there was no clear bias for downward microsaccades after the drift response (*Figure 2—figure supplement 4*).

## Discussion

Our results demonstrate that slow fixational drift eye movements are systematically influenced by visual transients. These results motivate much deeper neurophysiological investigation of drift eye movement control, especially given that these eye movements are incessant and modulate the spatio-temporal statistics of input images to the visual system (*Kuang et al., 2012*; *Rucci et al., 2007*; *Rucci and Victor, 2015*). Classic brainstem neurophysiological studies of the oculomotor system

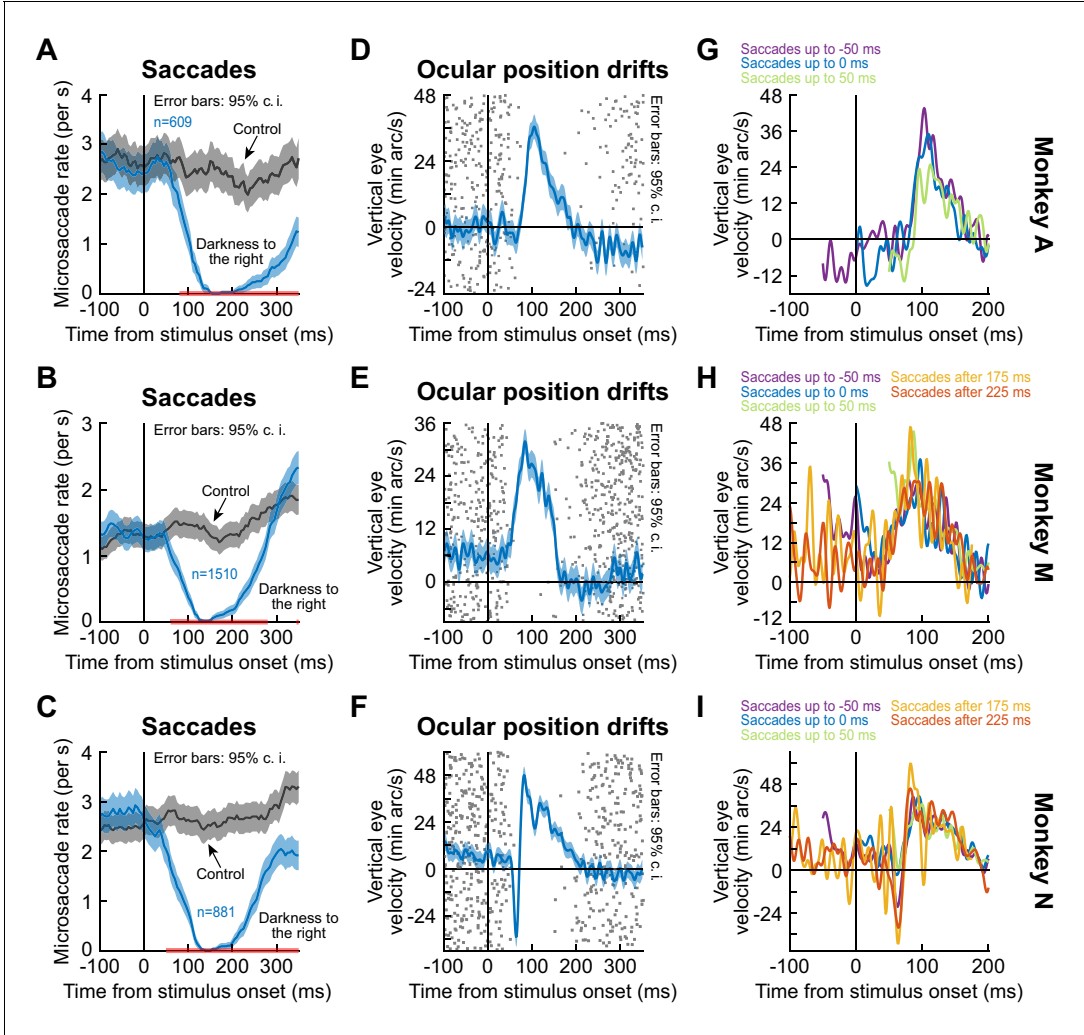

**Figure 4.** The drift response starts when saccades are inhibited. (A–C) Our monkeys exhibited known saccadic inhibition (*Buonocore et al., 2017a*; *Buonocore and McIntosh, 2008*; *Edelman and Xu, 2009*; *Hafed and Ignashchenkova, 2013*; *Hafed et al., 2011*; *Reingold and Stampe, 2002*) after stimulus onset (split view condition shown; *Figure 4—figure supplement 1* shows the fixation flash condition). Error bars denote 95% confidence intervals. (D–F) In the same sessions, saccade-free fixation epochs (no saccades from −100 to 350 ms) showed the ocular position drift response (error bars denote 95% confidence intervals). The timing of this drift response was linked to the timing of saccadic inhibition (individual dots show individual trial rasters of microsaccade onset times from A–C). The drift response started when microsaccades were inhibited. (G–I) When we combined (A–C) and (D–F), we found that even when microsaccades occurred near the time of drift response (i.e. before or after with shorter saccade-free epochs than in D–F), the drift response could still occur. In (G), two curves are missing because they had <10 trials due to the prolonged saccadic inhibition profile in (A). The different colors show traces in which there were microsaccades within a given time range relative to stimulus onset. For example, 'Saccades up to −50 ms' means that the curve shows average smooth eye velocity from −50 ms onwards, since earlier times had saccadic velocities in them. Similarly, the curve for 'Saccades after 175 ms' would mean that average eye velocity is shown up to 175 ms, since later times relative to stimulus onset had saccadic velocities in them. Error bars in (D–F) denote 95% confidence intervals. Also see *Figure 4—figure supplement 1* and *Figure 2—figure supplements 3* and *4*.

The online version of this article includes the following source data and figure supplement(s) for figure 4:

**Source data 1.** Excel table with the source data for this figure.

**Figure supplement 1.** The ocular position drift response occurred during saccadic inhibition, and it also occurred even with nearby saccades.

(e.g. reviewed in: *Takahashi and Shinoda, 2018*) are too coarse to provide the mechanistic insights necessary for explaining our observations. They need to be revisited with the perspective of understanding millisecond-by-millisecond tiny deviations of eye position during fixational drift.

Visual representations also need to be considered. Given the temporal transience of the drift response and its dependence on spatial frequency, it is likely that visual responses in oculomotor

control circuitry play a role. One possibility is that low spatial frequency stimuli increase the spatial uncertainty of intended gaze position. It is conceivable that such increased uncertainty, coupled with overrepresentation of the upper visual field in the SC (*Hafed and Chen, 2016*), as well as earlier visual bursts for upper visual field locations, results in transient upward drift responses. That is, increased spatial uncertainty increases the size of the active population of neurons (*Hafed and Krauzlis, 2008*), and if more neurons represent the upper visual field (*Hafed and Chen, 2016*), then the transient response would be biased in the upward direction. Indeed, the SC influences eye position (*Hafed et al., 2008*); what remains is to understand whether instantaneous SC activity landscapes, in cooperation with the deeper brainstem, can explain instantaneous variations in fixational eye position. This would suggest fairly high level central nervous system control of these eye movements.

Given the short latency of the drift response, an even more intriguing possibility than SC involvement is the contribution of downstream oculomotor control structures. The question of whether primate pre-motor and motor brainstem nuclei possess early, and feature-tuned, visual responses remains completely open. In the cat, visual responses in the nucleus raphe interpositus (rip) have been reported (*Evinger et al., 1982*), and there are hints of visual responses in monkey rip (*Busettini and Mays, 2003*; *Everling et al., 1998*; *Missal and Keller, 2002*), although no visual tuning was explicitly investigated. Given both the preponderance and richness of SC visual responses (e.g. *Chen and Hafed, 2018*; *Chen et al., 2019*; *Chen et al., 2018*), and given the fact that the SC projects to brainstem oculomotor control circuits, we hypothesize the presence of feature-tuned visual responses in such oculomotor control circuits. Such feature-tuned responses could be particularly useful to interpret the intriguing orientation tuning effects that we observed in *Figure 3B*. Our ongoing neurophysiological experiments do support this view (*Buonocore et al., 2020*), and they allow us to predict that lower brainstem visual responses might be additionally critical for the ocular position drift characteristics that we observed in the current study.

Naturally, one might also wonder whether the drift response is specific to monkeys. We think not. Indeed, in an earlier human study in which natural images (rich in low spatial frequency content) were presented during fixation, drift speeds were notably the highest in the very first analyzed time bin after image onset (*Poletti, 2010*). While the time bins in that study were too coarse to explore the short latency responses that we saw, these human results are, at least, consistent with our observation that image onsets can rapidly modulate ocular drifts. Other oculomotor phenomena are also very well preserved between monkeys and humans: some notable examples from our own prior work include ocular following responses (*Chen and Hafed, 2013*) as well as saccade-related implications of SC visual field asymmetries (*Grujic et al., 2018*; *Hafed and Chen, 2016*; *Hafed and Goffart, 2020*) and SC foveal magnification (*Chen et al., 2019*; *Hafed and Goffart, 2020*; *Willeke et al., 2019*).

Functionally, we believe that the timing of the drift response in relation to the timing of saccadic inhibition (*Figure 4*) might provide potential hints about the functional roles of the drift response for visual coding and perception. Saccadic inhibition is thought to reflect a resetting of ongoing saccadic rhythms to enable reacting (often with a subsequent eye movement) to the external stimulus arriving asynchronously to such rhythms (*Buonocore et al., 2017a*; *Hafed and Ignashchenkova, 2013*). Increasing behavioral work is also suggesting that the brief period of such saccadic inhibition additionally provides an important time window during which perceptual and cognitive processing of the incoming external stimulus could proceed (*Bompas et al., 2020*; *Buonocore et al., 2017b*; *Salinas and Stanford, 2018*). As part of this brief processing, under some situations like in our image patterns that caused the drift response, it might be important to modulate the visual statistics of the input images with drift eye movements. Therefore, the drift response (which introduces a coherent motion of certain speed) could serve this function, and this would suggest a short timescale version of the longer term modulations of how drifts are believed to modulate image statistics (e.g. to enhance edges and equalize spectral content of images; *Kuang et al., 2012*). Indeed, a quick increase in eye speed during the drift response would likely momentarily dampen the spatial frequency whitening effects of drift eye movements in their normal regime of slower speeds and smaller spatial spreads. Rather, lower spatial frequency power may benefit more from our observed rapid drift modulations, and this needs to be confirmed by calculating the frequency content of the visual input to the retina during the drift response period. Coupled with evidence that different subclasses of visual neurons (e.g. in primary visual cortex) will be differentially activated by different speeds and amplitudes of fixational eye movements (*Snodderly, 2016*), it is conceivable that the

drift response that we observed could functionally serve to preferentially activate the most suitable classes of visual neurons in cortex (and sub-cortex) for processing the kinds of images that we presented in our experiments (and that were also presented in: *Poletti, 2010*). Indeed, during smooth pursuit initiation (i.e. with very small eye velocities, albeit still faster than the ocular position drifts in our current study), when we used the same flash paradigms as in the current study, we found that the stimulus onsets consistently reduced smooth eye velocity rather than enhanced it (*Buonocore et al., 2019*). This suggests that, during fixation, flexibility to enhance smooth eye velocity is maintained, potentially to aid in visual processing.

In all, our results directly complement an increasing body of research on slow ocular position drifts during fixation, both on how they might be controlled as well as how they might be used for supporting vision. Early behavioral work focused on control, and even eventually invoked the term 'slow control' to describe these eye movements (as opposed to 'drifts'). This already suggested a kind of central nervous system control over these eye movements (*Epelboim and Kowler, 1993*; *Nachmias, 1961*). Similarly, vestibular ocular and other reflexes due to subtle head movements could result in small eye movements on the scale of ocular position drifts (*Ditchburn and Ginsborg, 1953*; *Poletti et al., 2015*; *Schor and Westall, 1984*; *Skavenski et al., 1979*; *Steinman and Collewijn, 1980*), and ocular tracking of very slow and small-amplitude motion trajectories is possible in both humans and monkeys (*Cunitz, 1970*; *Martins et al., 1985*; *Skinner et al., 2019*). Our results add to these observations the possibility of very rapid access to early sensory information by the oculomotor system, and they will hopefully motivate studies and models of how ocular position drifts can be generated on a millisecond-by-millisecond basis.

This complements other recent work investigating the visual and perceptual consequences of drifts. For example, in *Poletti et al., 2010*, it was suggested that the brain has some internal knowledge of these movements, again consistent with central control. And, in theoretical accounts, drifts have been suggested to modulate image statistics for aiding in visual image processing (*Kuang et al., 2012*; *Rucci and Victor, 2015*). Interestingly, such accounts can still be functional if drifts were completely random movements, as long as these random movements possessed statistics that could functionally whiten the spectral content of natural images. Our results complement these latest theoretical accounts by demonstrating robust drift modulations on much shorter time scales, and which are also anything but random. A strong implication of our work is, then, that a random walk model of ocular position drifts (*Burak et al., 2010*; *Engbert and Kliegl, 2004*; *Engbert et al., 2011*; *Herrmann et al., 2017*; *Kuang et al., 2012*) is not necessarily accurate.

## Materials and methods

### Animal preparation and laboratory setup

Monkey N was 12 years old at the time of the experiments; monkeys A and M were 6–9 years old. Each monkey was initially implanted with a titanium head holder under aseptic surgical conditions, as described earlier (*Chen and Hafed, 2013*). In a subsequent procedure, each monkey was additionally implanted with a scleral search coil in one eye (left for monkey A and right for monkeys N and M), in order to allow precise eye tracking using the magnetic induction technique (*Fuchs and Robinson, 1966*; *Judge et al., 1980*). For some experiments in monkey M (e.g. *Figure 2*), we also tracked eye movements binocularly. In this case, we performed an additional procedure to implant a search coil in this monkey's other eye. Eye coil implant procedures in our laboratory were described in detail earlier (*Chen and Hafed, 2013*).

The laboratory setup consisted of a cathode ray tube (CRT) display placed in front of the monkey in an otherwise dim room. The display subtended approximately + / - 15 deg horizontally and + / - 11 deg vertically when the monkey fixated at its center, and it had a 120 Hz refresh rate. The monkey's head was surrounded by a cube generating electromagnetic fields for eye tracking (Remmel Labs). When we performed pupillometry (e.g. *Figure 1—figure supplement 2*), we placed a video-based eye tracking camera (EyeLink 1000, desktop mount; SR-Research, Ontario, Canada) in front of the monkey under the CRT display. The eye tracking software output eye positions and pupil diameters (of the right eye) for synchronization with our existing real-time experiment control system, described in *Chen and Hafed, 2013*; *Tian et al., 2016*.

All experiments involved initially fixating a white fixation spot over a gray background. The luminance of the gray background was 29.7 cd/m². The luminance of the white fixation spot was 86 cd/m². When we presented black stimuli, the luminance was <0.02 cd/m². For all other stimuli (e.g. spatial frequencies), the details of the stimuli are provided below. Display resolution (~34–36 pixels/deg) was always higher than the Nyquist limit associated with the highest spatial frequency stimulus that we tested, and the display was linearized and calibrated before the experiments.

The monkeys were also all highly trained on fixation tasks that exercised their abilities for accurate and precise fixation of a small target. For example, monkey N previously contributed to gaze-contingent experiments in which forced foveal motor errors of only a few minutes of arc could be corrected for by the monkey's eye position (*Tian et al., 2016*; *Tian et al., 2018*), and monkeys M and A successfully used smooth tracking of very slow motion trajectories, demonstrating that they could control their smooth eye movement system at speeds similar to or even lower than the speeds of ocular drift eye movements during fixation (*Skinner et al., 2019*).

## Behavioral tasks

Experiment one was called the split view stimulus paradigm. The monkey fixated a small white spot (approximately 5 × 5 min arc) presented over a gray background (*Chen and Hafed, 2013*). After an initial fixation interval (approximately 500–1400 ms), a stimulus onset occurred, which had the following properties. The stimulus onset was marked by turning one entire half of the display (right or left, randomly picked across trials) to black until the end of the trial (approximately 500–2000 ms after stimulus onset). The vertical edge associated with such a split view stimulus was gaze-contingently (*Chen and Hafed, 2013*; *Tian et al., 2016*; *Tian et al., 2018*) updated, such that the edge between the light and dark regions of the retinal image was experimentally stabilized in real time relative to the fovea. This gaze-contingent image manipulation was relevant for other aspects of the task that are beyond the scope of the current analyses, especially because the current study focuses on time periods that are too short-lived to be affected by whether we used retinal image stabilization of the split view stimulus or not. To maintain visibility of the fixation spot despite the half-display of darkness that went right down its middle, we additionally enforced a kind of foveal sparing of the split view stimulus. Specifically, for a circle of radius 0.74 deg centered on instantaneous gaze position, we maintained background luminance even for the half of the circle that resided in the darkened half of the display (i.e. the circle was not split into a dark and bright half like the rest of the split view display). The fixation spot was always visible and always stable in the display (i.e. did not move gaze-contingently). We also interleaved control trials in this experiment that had identical timing; however, in this case, there was never a stimulus onset on the display (i.e. the monkey simply fixated steadily until trial end with no visual transients at all). We analyzed 5193, 2156, and 3860 trials in this experiment from monkeys M, A, and N, respectively. Typically, we collected 150–330 trials per session and repeated the experiment across multiple sessions.

Experiment two was called the black fixation flash experiment. Each monkey fixated a white spot over a gray background like in Experiment one. After an initial fixation interval (approximately 550–1800 ms after initiating fixation), the entire display turned black for one single display frame (~8 ms duration given our 120 Hz refresh rate) before returning to its original state. Therefore, for one display frame, the fixation spot was rendered momentarily invisible. However, such a short duration of fixation spot occlusion is not expected to affect the eye movements that we analyzed (e.g. the drift response that we observed in this experiment was similar to that in Experiment one with a continuously visible fixation spot; see Results). In other control conditions in this experiment, we interleaved trials in which either no flash occurred at all, or in which the single frame flash was now a flash of a localized square of 1 × 1 deg dimensions centered at 2.1 deg horizontally either to the right or left of the fixation spot. The flashed square was also black, and the fixation spot was still visible in this case (because the flash did not occlude it). This experiment was similar to one we used recently under smooth pursuit conditions (*Buonocore et al., 2019*). We analyzed 1671 and 1837 trials from monkeys M and A, respectively, in this experiment.

In our third experiment, we replicated Experiment two but with white rather than black flashes (the white fixation flash paradigm). We analyzed 5191 and 3112 trials from monkeys M and A, respectively, in this experiment.

Finally, Experiments four and five tested spatial frequency and orientation tuning, respectively, of the ocular position drift response in monkey M. We repeated the fixation flash paradigms described

above. However, instead of full-screen black or white flashes, we presented a full-screen grating of a given stimulus property for approximately 260–700 ms. Across trials, we varied the spatial frequency (Experiment four) or the orientation (Experiment five) of the presented grating. There were no control or localized flash trials in these new experiments. Instead, we exploited the available numbers of trials to collect by varying the stimulus properties being tested across trials. For spatial frequency tuning, we tested five different spatial frequencies across trials (0.55, 1.1, 2.2, 4.2, and 6.8 cycles/deg; cpd). For orientation tuning, we tested four different orientations (vertical, 45 deg clockwise from vertical, horizontal, and 45 deg counterclockwise from vertical). Grating contrast was always maximal (100%). Also, across trials, we picked from two different grating phases (one in which the grating was white at the center of the display, near the fixation spot, and one in which the grating was black at the center of the display; that is a phase difference of $\pi$ from the other condition). We analyzed 9967 trials from the spatial frequency manipulation and 9856 trials from the orientation manipulation.

To test for the relationship between the ocular position drift response and pupillary constrictions associated with stimulus onset, we measured pupil diameter in monkey A (Experiment six). We ran the monkey on a variant of the fixation flash tasks that we knew would evoke a drift response (i.e. a full-screen flash of a visual pattern consisting of a high-contrast vertical grating). We confirmed such a drift response by measuring eye position using a scleral search coil. Simultaneously, we analyzed pupil diameter using a video-based eye tracker, and we analyzed variations in pupil diameter as a function of time from stimulus onset. This allowed us to relate the pupillary constriction response after stimulus onset to the ocular position drift response that we report in this study. Note that the video-based eye tracker was not suitable to detect the drift response in the same eye for which we measured the pupil. That is why we tracked the other eye with a scleral search coil. That is, we exploited our observation that the drift response is a binocular phenomenon (*Figure 2*). We could have measured the pupil diameter in the coil eye, but our initial intent was to measure the drift response binocularly (with a coil in one eye and a video-based eye tracker for the other). However, it quickly became clear that the video-based eye tracker was simply not suitable for properly measuring the drift response, consistent with prior observations in the literature (*Choe et al., 2016*; *Drewes et al., 2014*; *Hooge et al., 2019*; *Kimmel et al., 2012*; *Wyatt, 2010*). We analyzed 6203 trials from this comparison.

To investigate whether the drift response was associated with a simultaneous head movement, we also repeated Experiment two (with only the full-screen flash condition) on monkey M while simultaneously recording eye and head position. Eye position recording (left eye) was identical to the procedures described above. For head position measurements, we placed a search coil (identical to the coils that we implanted in the monkeys' eyes) at one of three possible positions on the head (e.g. attached to a recording chamber implanted on the head or attached to the titanium head holder screwed on the monkey's skull, and so on). The head coil had lower resistance (40 ohms) than the eye coils (typically around 90 ohms), resulting in an even more sensitive measure of potential head movements. We used three positions of head coils, across three different sets of experiments, to make sure that we could catch all possible axes of head motion that could occur in our experiments. Specifically, the magnetic fields that were generated around the monkey's head were optimized for measuring horizontal and vertical eye position (i.e. with a nominal orientation of the eye coil within the magnetic fields). However, with head coil placement, positions of recording chambers on the head or the titanium head holder screwed on the skull resulted in non-standard orientations of the head coil relative to the magnetic fields. This meant that some axes of potential head movements could be missed with only one inopportunely chosen head coil position. As a result, we varied head coil position on the head, across three different experiments, in order to be absolutely sure that our drift response could not be explained by a reflexive head movement after stimulus onset. We digitized the head coil output using the same amplification and acquisition system as our eye coil outputs.

To obtain an approximate calibration of head coil measurements, we brought the monkey chair to the laboratory without a monkey, and we then placed the head coil at one position relative to the head holder (which is always at a fixed position relative to both the chair and the monkey's head). We then measured calibration output by physically moving the head coil by 3 mm in different directions from its standard position. At each position, we recorded 5000 ms of data, which we then averaged. This gave the approximate calibration values in *Figure 2—figure supplement 1*.

## Data analyses

We detected saccades and microsaccades using established methods in our laboratory (*Bellet et al., 2019*; *Chen and Hafed, 2013*). We also manually inspected all trials to ensure correct detection, and to also remove blinks.

To analyze our drift responses, we considered only trials in which there were no saccadic or blink events within an interval starting from −100 ms relative to stimulus onset and ending at +200 ms relative to stimulus onset. In additional control analyses, we extended this so-called saccade-free interval forward in time in steps of 50 ms until +350 ms after stimulus onset. That is, in the longest analysis interval that we considered, there were no saccadic or blink events between −100 ms and 350 ms from stimulus onset.

To absolutely ensure that we excluded any possibility of saccadic eye speeds (for even the smallest microsaccades) that might bias our measurements of average eye position across trials, we added one extra step after saccade detection, just as a sanity check: we inspected all accepted trials based on saccade detection and the above interval, and we ensured that eye velocity profiles did not have any clear outliers above the levels expected from ocular position drifts and measurement noise for a given session and monkey. This ensured that even rare electronic artifacts (lasting for only 1 ms or so) in scleral search coil measurements were not corrupting our ocular position drift interpretations.

We summarized the drift response properties by averaging eye position or velocity traces for a given stimulus type. When averaging eye positions, we aligned traces horizontally on stimulus onset time and vertically on the eye position that existed at this time. This way, we could observe the drift responses regardless of the absolute eye position that existed at trial onset. In other analyses (e.g. *Figure 2—figure supplement 3*), we explicitly considered the effect of absolute eye position. Specifically, in each session, we estimated the trial-to-trial variance of eye position (due to fixational eye movements) at the time of stimulus onset (we averaged the final 50 ms of eye position data right before stimulus onset in each trial, and the across-trial distribution provided an estimate of fixation variability). Then, we considered only trials above the median vertical eye position across the population or only trials below the median vertical eye position. We then pooled the above-median and below-median populations across sessions to increase the numbers of observations (and, therefore, the statistical confidence of our results). For presentation in *Figure 2—figure supplement 3* of these median split data, we still aligned the eye positions at the time of stimulus onset as we did for the other figures (see above). This allowed us to directly demonstrate that the drift response was the same in timing and amplitude whether the starting eye position was below or above median eye position.

For analyzing microsaccades, we estimated microsaccade rate as a function of time from stimulus onset using similar methods to our earlier work (*Buonocore et al., 2017a*; *Hafed et al., 2011*). Briefly, we used a running window of width 80 ms, during which we estimated rate as the average number of movements observed per time window (and normalized to a unit of movements per second) (*Hafed et al., 2011*). We then stepped the window forward in time, in steps of 5 ms, starting from −300 ms and ending at +600 ms relative to stimulus onset.

We also checked whether the drift response happened when a nearby microsaccade still occurred (e.g. *Figure 4G–I*). To do this, we binned trials based on the timing of microsaccades. For example, we looked for trials in which there was a microsaccade (or more) occurring up to a time of, say, +50 ms after stimulus onset. Then, we plotted saccade-free eye velocity after this time to look for the drift response. We used a similar procedure for other times earlier than +50 ms. For microsaccades after the drift response, we also did the same. For example, we checked if there was a microsaccade (or more) starting from a time of, say, +225 ms after stimulus onset (i.e. after the ocular drift response). In this case, the drift response occurred earlier, so we plotted saccade-free eye velocity up to the time of, say, +225 ms.

To further relate microsaccades to the ocular drift response, we investigated whether the first microsaccade after the drift response was biased downwards in direction (e.g. to compensate for the predominantly upward drift response). We used the following procedure. In all of our saccade-free trials in the standard interval of −100 ms to +200 ms relative to stimulus onset, we searched for the first microsaccade to occur after this interval. We then plotted the angular distribution of this first microsaccade and related it to the control data without any stimulus onsets, but with the same

analysis workflow of saccade-free fixation between −100 ms and +200 ms (*Figure 2—figure supplement 4*).

For pupil diameter measurements, our data acquisition system output analog voltages proportional to pupil diameter values calculated by the video-based eye tracker. We stored digitized measurements of these voltages in synchrony with our stimulus events, and that is why we reported these measurements using arbitrary units in *Figure 1—figure supplement 2*. We used the average pupil diameter in the final 100 ms before stimulus onset in a given trial as the baseline measurement. We subtracted this baseline measurement from all pupil measurements within the trial. Then, we averaged across trials. This allowed us to display the relative change in pupil diameter after stimulus onset (again, in arbitrary units). We ensured that we analyzed saccade-free epochs of pupil diameter, like we analyzed the ocular position drift responses in all other experiments. We measured saccades in both the video-based eye tracker signal as well as the other eye, which was tracked with a scleral search coil. This way, our analyses of pupil diameter dynamics after stimulus onset were comparable to the analyses of saccade-free ocular position drift responses that we report in this study. For the linear fits in *Figure 1—figure supplement 2*, we used a piece-wise linear model to estimate the onset of the smooth ocular drift response or the onset of the pupil constriction. This approach is the same as that used for smooth pursuit eye movements earlier (*Krauzlis and Miles, 1996*).

Because we were dealing with very subtle eye position effects even in no-stimulus controls (e.g. *Figure 2A*), we first confirmed that potential electronic drifts for our scleral search coil system cannot explain our results, and we did this before performing any of the analyses above. Specifically, sometimes, scleral search coil systems might exhibit a subtle drift in eye position calibration during the course of a given session (i.e. across tens of minutes or even hours). To confirm that such a potential technical artifact was too slow (if it existed at all) to account for even the baseline no-stimulus drift pattern of a given monkey in our data (e.g. *Figure 2A*), we compared such baseline drift speed (which we deemed physiological) to any potential electronic drifts in eye tracker calibration across trials within a session. To do this, we measured average eye position in the final 30 ms before stimulus onset (including sham stimulus onset in control conditions). We then plotted this measurement as a function of trial number. Given the trial durations, this allowed us to estimate an 'electronic drift speed' measure. When it existed, this electronic drift speed was at least an order of magnitude slower than baseline no-stimulus drift speeds of our monkeys. We are, therefore, confident that our results are physiological drifts of eye position, and not measurement artifacts.

Finally, for the simultaneous eye and head position measurements, we analyzed eye position and velocity as described above, including our measures to ensure saccade-free epochs for analyzing the drift response. For head position, we obtained the output of each channel of the head coil, and we also obtained a differential of it using the same smoothing differentiation filter that we used to obtain eye velocity. We visualized the position measurements in *Figure 2—figure supplement 1* to confirm that the drift response was not accompanied by a head response (the head data in the figure are very similar in variance to the measurements obtained during head coil calibration with no monkey at all; that is at the level of electronic noise of the system). Head movement data were analyzed from the very same trials that we included for the eye movement analyses.

## Statistics

We used descriptive statistics throughout the study. All figures show and define error bars. We also did not show any average curves having fewer than 10 repetitions, because such few repetitions were deemed unreliable; this is the reason for the missing two curves in *Figure 4G* in relation to the similar panels of *Figure 4H,I* (saccades were simply too rare due to the saccadic inhibition shown in *Figure 4A*). Typically, we had many more numbers of repetitions than 10, as shown in the insets of the figures.

To estimate temporal parameters such as the onset time of an ocular drift response, we compared measurements on trials with a stimulus onset to corresponding control trials without any stimulus onset. For each of the stimulus or control populations of repetitions, we estimated the 95% confidence intervals surrounding the mean measurement. We then marked all time points in which such 95% confidence intervals of the two compared curves did not overlap. This was done using pink horizontal lines on all relevant x-axes in all figures. Due to noise (particularly in differentiated eye velocity measurements), there could be individual milliseconds or pairs of milliseconds in which there was no overlap between the 95% confidence intervals. We still displayed these time points in

all figures. However, we consider a significant difference as one in which the 95% confidence intervals do not overlap for at least 20 consecutive milliseconds.

We used the above procedures on either eye position or eye velocity independently, and we include results in all figures for both eye position and eye velocity curves. Because eye velocity computations necessarily involve some smoothing operation to regularize the noise associated with numerical differentiation, they introduce some temporal blurring of signals (both backwards and forwards in time). On the other hand, lack of overlap in the 95% confidence intervals of position traces would be too conservative, because it would estimate a later onset of the drift response (velocity is more sensitive than position for detecting onset of smooth eye movements). We therefore elected to show in the figures all statistical comparisons independently for both eye position and eye velocity traces, for the sake of completeness. In this regard, the position results can be considered to be conservative estimates of drift response onset time, whereas the velocity results can be considered to be slightly more liberal estimates. Numbers (e.g. p-values) reported in the text were based on eye velocity statistics, because eye velocity also allowed estimating the duration of the drift response (the end of which being when the eye velocity curve returned back to baseline with overlapping 95% confidence intervals).

In some cases, we additionally statistically compared drift responses across conditions (e.g. for right versus left eye). To do this, we defined a measurement interval based on the timing of the drift response that we observed (50–140 ms in black or white fixation flash paradigms and 80–140 ms in the spatial frequency and orientation paradigms). During this interval, we searched for peak eye velocity and characterized its magnitude and latency. When comparing two measurements (e.g. right versus left eye peak velocity latency), we used non-parametric Wilcoxon rank sum tests. When comparing multiple conditions (e.g. five different spatial frequencies), we used non-parametric Kruskal-Wallis tests. We used non-parametric tests because our measurements were not normally distributed. Finally, when we reported 'adjusted p-values' in Results, we were reporting corrections for multiple comparisons using Bonferroni methods.

For confirming a lack of head movement around the time of the drift response (*Figure 2—figure supplement 1*), we obtained a paired measurement of eye or head velocity from each trial of a given experiment. The first measurement of the pair was average eye or head velocity in the interval ($-100$–0 ms from stimulus onset), and it served as a baseline velocity. The second measurement of the pair was average eye or head velocity in the interval (50–150 ms from stimulus onset), chosen to coincide with the time of the expected drift response. For a given head coil position, we then performed a Wilcoxon rank sum test on vertical eye velocity across both intervals. If there was a drift response, then the vertical eye velocity in the second interval was expected to be statistically significantly different from the vertical eye velocity in the second interval (*Figure 2—figure supplement 1A*). We then performed the Wilcoxon rank sum test on each channel of head velocity (*Figure 2—figure supplement 1B,C*). If the drift response was accompanied by a head movement, then there should have been a statistically significant change in head velocity (in either channel or both) after stimulus onset when compared to before stimulus onset. Note that we could have also performed the statistics directly on the raw position measurements, and we would have reached similar conclusions (*Figure 2—figure supplement 1*). However, since the monkey had a slight baseline drift even without any stimulus onset, statistics on eye position before and after stimulus onset would have been significant (for eye position) even if there was no drift response at all. We therefore used velocity because it effectively detrends the eye position signal. We also analyzed each channel of head coil output individually because head coil orientation was not always perfectly orthogonal to the inducing magnetic field in the laboratory (see above).

## Acknowledgements

We were funded by the Deutsche Forschungsgemeinschaft (DFG) through the Research Unit: FOR 1847 (project A6: HA6749/2-1). We were also funded by the Werner Reichardt Centre for Integrative Neuroscience (CIN; DFG EXC307) and the Hertie Institute for Clinical Brain Research. TM and ZMH were additionally supported by a CIN intramural grant (Mini_GK 2017–04). We thank Tong Zhang and Matthias Baumann for helping us to perform the control experiments measuring both eye and head position simultaneously, and we also thank Joachim Bellet for help in collecting monkey N data.

## Additional information

### Funding

| Funder | Grant reference number | Author |
| --- | --- | --- |
| Deutsche Forschungsgemeinschaft | HA6749/2-1 | Antimo Buonocore<br>Ziad M Hafed |
| Deutsche Forschungsgemeinschaft | EXC307 | Tatiana Malevich<br>Antimo Buonocore<br>Ziad M Hafed |
| Werner Reichardt Centre for Integrative Neuroscience | Mini_GK 2017–04 | Tatiana Malevich<br>Ziad M Hafed |

The funders had no role in study design, data collection and interpretation, or the decision to submit the work for publication.

### Author contributions

Tatiana Malevich, Antimo Buonocore, Conceptualization, Data curation, Formal analysis, Validation, Visualization, Writing - original draft, Writing - review and editing; Ziad M Hafed, Conceptualization, Data curation, Formal analysis, Supervision, Funding acquisition, Validation, Investigation, Visualization, Methodology, Writing - original draft, Project administration, Writing - review and editing

### Author ORCIDs

Tatiana Malevich (iD) https://orcid.org/0000-0003-3928-6248
Antimo Buonocore (iD) https://orcid.org/0000-0003-3917-510X
Ziad M Hafed (iD) https://orcid.org/0000-0001-9968-119X

### Ethics

Animal experimentation: We tracked eye movements in 3 male rhesus macaque monkeys trained on behavioral eye movement tasks under head-stabilized conditions. The experiments were part of a larger neurophysiological investigation in the laboratory. All procedures and behavioral paradigms were approved (CIN3/13 and CIN4/19G) by ethics committees at the Regierungspräsidium Tübingen, and they complied with European Union directives on animal research.

### Decision letter and Author response

Decision letter https://doi.org/10.7554/eLife.57595.sa1
Author response https://doi.org/10.7554/eLife.57595.sa2

## Additional files

### Supplementary files

• Transparent reporting form

### Data availability

All data generated or analyzed during this study are included in the manuscript and supporting files. Source data for figures are included.

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
