## [Decision Letter]

**Acceptance summary:**

This study is a highly rigorous investigation of ocular drift, i.e., of the tiny eye movements that take place when we attempt to stare at a visual target. Prior work suggested that, although such drift can enhance vision in some ways, the motion itself had little structure and thus could be characterized as a random walk. Contrary to this simplification, here the authors show that a rapid, stereotyped, and reliable drift response is observed when specific visual stimuli are shown, demonstrating that such eye movements are carefully controlled by the nervous system.

**Decision letter after peer review:**

Thank you for submitting your article "Rapid stimulus-driven modulation of slow ocular position drifts" for consideration by *eLife*. Your article has been reviewed by three peer reviewers, one of whom is a member of our Board of Reviewing Editors, and the evaluation has been overseen by Timothy Behrens as the Senior Editor. The following individual involved in review of your submission has agreed to reveal their identity: Martina Poletti (Reviewer #2).

The reviewers have discussed the reviews with one another and the Reviewing Editor has drafted this decision to help you prepare a revised submission.

Summary:

This study presents a set of comprehensive measurements of ocular drift, the drift in eye position that occurs while subjects are supposed to be fixating on a given visual target. The results are interesting because they reveal that, far from a random walk or diffusive wandering around the intended fixation target, the slow, tiny drift exhibits a number of regularities: it shows a short-latency upward bias after the onset of "large" visual stimuli, the response is binocular but unrelated to vergence, the strength of the drift varies with certain stimulus properties, such as spatial frequency and orientation, and, to some degree, it seems to be coordinated over time with other oculomotor events, such as microsaccades. All of this was robust and repeatable across monkeys. All reviewers thought the work was carefully done and clearly described.

Essential revisions:

The reviewers had two main reservations about the work, both of which were considered critical.

1) The novelty of the results. There is conceptual overlap with previous work in humans that is barely touched upon. This is detailed in the comments of reviewer 2, but was a concern for all reviewers. Overall, the authors must do a better job citing the literature and acknowledge previous work done on ocular drift, and explain what their study is adding and how it differs compared to what we already know on this topic.

2) The possibility of an artifact. A key finding is that the modulation of drift occurs on a short time scale and has a sharp onset, particularly for stimuli with low spatial frequency. This would imply that whatever mechanism controls ocular drift must have the ability to modulate the speed of the eye very quickly. This would be quite remarkable and, in fact, raises the question of whether the effect is the result of an artifact. Although the authors used high precision techniques, artifacts due to tiny head movements could potentially explain the results. This is detailed in the comments of reviewer 3. This is even more of a concern given previous evidence that ocular drift compensates for head fixational motion.

Reviewer #1:

This manuscript presents careful measurements of ocular drift to reveal quick stereotypical responses to certain stimuli, as well as other regularities in such movements. The data are pretty cool in that the time scale of the effects is faster than expected. They also depend on very careful measurement of very small movements, and suggest that the oculomotor system coordinates its activity with an exquisite, perhaps under-appreciated level of spatial and temporal precision.

One obvious weakness is that the work does not provide any direct clue, or even some discussion, as to the potential functional significance of these regularities. Other work has shown that drift is functional for fine spatial vision (Rucci and Victor, 2015), and that it compensates for microscopic head movements (Poletti et al., 2015), for instance. Some discussion of the functional implications of the results reported here is warranted.

Reviewer #2:

This is a really interesting work conducted rigorously and clearly written. The authors show that ocular drift, the incessant jittery motion of the eye during fixation, is modulated, on a short time scale, by visual transients. This is an important conclusion because little is known on the mechanisms controlling ocular drift and, according to the dominant view, ocular drift is a purely random process over which no control is exerted by the oculomotor system.

1) Although it is true that ocular drift is often seen as a random process, there has been research showing a certain degree of control over ocular drift (see Nachmias, 1961; Steinman et al., 1973; Epelboim and Kowler, 1993; and more recently, Poletti et al., 2010 JN; Poletti et al., 2015; Shelchkova et al., 2019 and Intoy and Rucci, 2020). Some of these studies showed that the span and direction of drift change based on the task and the visual stimulus.

It is important that the authors emphasize how their findings differ from this previous work. In particular, while the modulations of ocular drift described in these papers occur over a longer time scale, here, the authors show sharp modulations over a very short temporal scale and their spatial modulations are in the order of just a couple of arcminutes.

2) The authors should speculate whether the modulation reported here is something that is likely to occur in humans as well. It is interesting that Poletti and Rucci, 2010, reported a transient increase in human ocular drift speed during the first hundreds of milliseconds after the abrupt onset of a full screen natural stimulus with a significant presence of low spatial frequencies. A result that seems to support the idea that the findings reported here extends to humans.

3) Figure 1. Please use regular y axis rather than a scale bar for the vertical dimension in the graphs. Scale bars make it really cumbersome to interpret the data. I would also be consistent across all the figures and use 95% error bars since your graphs are based on a really high number of trials.

4) A peak speed of 33-45 '/s seems rather low for ocular drift, where the authors filtering their eye movement data? Or maybe their selection criteria were such that only lower speed drifts were selected for analysis?

5) Most of the discussion focuses on the neurophysiological mechanisms controlling ocular drift. However, even if short and small, the drift modulation described here may have an impact on, and possibly it may be functional for, visual perception. I would include in the discussion some speculation about the effects of this drift modulation higher up in the visual processing pipeline (e.g., in V1, see for example Snodderly, 2016 on the effects of ocular drift in V1), and about the consequences of this modulation for visual perception.

Reviewer #3:

This is a technically strong and thorough study of small-scale drift eye movements in macaque monkeys and it relation with visual input. The manuscript is well written, and the analyses are appropriate. The main conclusion is that drift is not random but depends on the visual input. Except for major concern #1 the data support this conclusion.

Given previously published studies from Michele Rucci's group in collaboration with Jonathan Victor, however, this is not an overly surprising finding. In fact, those publications went substantially further by showing both theoretically and experimentally that drift improves visual function (and not just that visual input affects drift).

Major concern:

1) As stated above, the data are strong and the authors have addressed most potential confounds. There is one, however, that appears to be missing. When suddenly exposed to a flash, it could well be that the animals head moves slightly (presumably backward) within the search coil frame. This could cause artefactual position changes due to slight inhomogeneities in the magnetic field. The stimulus dependence that the authors find would then be attributed to the potential to "scare" the animal. For most applications magnetic field homogeneity is good enough, and a head-holder solid enough, but given that we're looking at such small changes in inferred eye position, this remains a possibility that should be addressed. Measurement of head position (using a separate head coil) could address this relatively easily.

---

## [Author Response]

Essential revisions:The reviewers had two main reservations about the work, both of which were considered critical.1) The novelty of the results. There is conceptual overlap with previous work in humans that is barely touched upon. This is detailed in the comments of reviewer 2, but was a concern for all reviewers. Overall, the authors must do a better job citing the literature and acknowledge previous work done on ocular drift, and explain what their study is adding and how it differs compared to what we already know on this topic.

We have now significantly expanded our Introduction and Discussion sections to address this point. More specific responses are provided below in the responses to the individual reviewers. Briefly, there has been no prior demonstration of such rapid, stimulus-driven modulation of ocular position drift eye movements as we report here. The novelty and implications of this finding are, in our opinion, quite strong, particularly for our understanding of the neurophysiological mechanisms for precisely controlling eye position.

2) The possibility of an artifact. A key finding is that the modulation of drift occurs on a short time scale and has a sharp onset, particularly for stimuli with low spatial frequency. This would imply that whatever mechanism controls ocular drift must have the ability to modulate the speed of the eye very quickly. This would be quite remarkable and, in fact, raises the question of whether the effect is the result of an artifact. Although the authors used high precision techniques, artifacts due to tiny head movements could potentially explain the results. This is detailed in the comments of reviewer 3. This is even more of a concern given previous evidence that ocular drift compensates for head fixational motion.

We have now added three new sets of experiments in which we explicitly measured head position simultaneously with eye position (please see the new Figure 2—figure supplement 1 in the revised manuscript and its associated text). More details are provided below in the responses to reviewer 3. Briefly, we are very confident that our modulations of ocular position drifts are independent of subtle head movements.

Reviewer #1:This manuscript presents careful measurements of ocular drift to reveal quick stereotypical responses to certain stimuli, as well as other regularities in such movements. The data are pretty cool in that the time scale of the effects is faster than expected. They also depend on very careful measurement of very small movements, and suggest that the oculomotor system coordinates its activity with an exquisite, perhaps under-appreciated level of spatial and temporal precision.

Thank you very much. We wholeheartedly agree. We also believe that our results can help us to better understand the neurophysiological mechanisms (e.g. in the brainstem) for precisely controlling eye position. Our ongoing neurophysiology experiments in the laboratory are directly motivated by the current manuscript.

One obvious weakness is that the work does not provide any direct clue, or even some discussion, as to the potential functional significance of these regularities. Other work has shown that drift is functional for fine spatial vision (Rucci and Victor, 2015), and that it compensates for microscopic head movements (Poletti et al., 2015), for instance. Some discussion of the functional implications of the results reported here is warranted.

We have now significantly expanded our Introduction and Discussion sections to better place our work in the literature and highlight its novelty (also to address the other reviewers’ comments). We have also discussed potential functional implications of our observations.

We have generally placed more emphasis on neurophysiological implications, since this is why we used monkeys in the first place. Also, the most intriguing aspect of our manuscript is that it describes a new discovery that we strongly feel is important to document, even if we still do not yet have a full grasp on its complete functional implications.

Reviewer #2:This is a really interesting work conducted rigorously and clearly written. The authors show that ocular drift, the incessant jittery motion of the eye during fixation, is modulated, on a short time scale, by visual transients. This is an important conclusion because little is known on the mechanisms controlling ocular drift and, according to the dominant view, ocular drift is a purely random process over which no control is exerted by the oculomotor system.1) Although it is true that ocular drift is often seen as a random process, there has been research showing a certain degree of control over ocular drift (see Nachmias, 1961; Steinman et al., 1973; Epelboim and Kowler, 1993; and more recently, Poletti et al., 2010 JN; Poletti et al., 2015; Shelchkova et al., 2019 and Intoy and Rucci, 2020). Some of these studies showed that the span and direction of drift change based on the task and the visual stimulus.It is important that the authors emphasize how their findings differ from this previous work. In particular, while the modulations of ocular drift described in these papers occur over a longer time scale, here, the authors show sharp modulations over a very short temporal scale and their spatial modulations are in the order of just a couple of arcminutes.

We agree that our original text was probably too brief in the Introduction and Discussion sections. This meant that even though we did already cite some of the same key papers that you mentioned here (e.g. Nachmias, 1961; Steinman et al., 1973; Intoy and Rucci, 2020) and others, some additional important citations were unfortunately missing. We have now remedied this issue in our revised manuscript (e.g. please see the significantly expanded Introduction and Discussion sections). We have also clarified how our new results strongly complement the existing literature in a very intriguing manner, exactly as you suggested.

2) The authors should speculate whether the modulation reported here is something that is likely to occur in humans as well. It is interesting that Poletti and Rucci, 2010, reported a transient increase in human ocular drift speed during the first hundreds of milliseconds after the abrupt onset of a full screen natural stimulus with a significant presence of low spatial frequencies. A result that seems to support the idea that the findings reported here extends to humans.

We suspect that this phenomenon will also happen in humans, and we now do remember this particular result that you referred to in Poletti and Rucci, 2010. Indeed, it may be possible that the first time bin studied in that paper included a component of our reported drift response. We have now mentioned this explicitly in our revised manuscript, especially because it demonstrates that it is quite likely that our observations would indeed generalize to humans. Please see Discussion, paragraph five. Thank you for pointing this out.

3) Figure 1. Please use regular y axis rather than a scale bar for the vertical dimension in the graphs. Scale bars make it really cumbersome to interpret the data. I would also be consistent across all the figures and use 95% error bars since your graphs are based on a really high number of trials.

We have now followed your advice and included full axes in all figures. The only exceptions were the individual panels showing example individual trials (e.g. in Figure 1 and Figure 1—figure supplement 1). We felt that it was less critical to include the full y axes for these panels with individual trials, especially given that we jittered the y positions of the individual curves to facilitate viewing of the eye movement trajectories. We hope that you are ok with this tiny exception from your overall advice (all other figures now follow your advice of showing full axes).

We have also now followed your advice on showing 95% confidence intervals as error bars in all of our figures. This resulted in much more consistent presentation of our results across all figures.

4) A peak speed of 33-45 '/s seems rather low for ocular drift, were the authors filtering their eye movement data? Or maybe their selection criteria were such that only lower speed drifts were selected for analysis?

We did not filter our digitized eye movement data for analyses. The digitized eye position traces were numerically differentiated to obtain speed estimates. Naturally, our differentiating filter (to obtain eye speed) included a built-in low-pass component in its filter response equation, but this is common best practice with numerical differentiation anyway. We suspect that the potentially low values of drift speed that you refer to relate to the fact that our monkeys were highly trained on fixating tiny visual targets. We believe that this is consistent with your earlier work (Cherici et al., 2012) comparing fixational precision across different individuals with various amounts of training to fixate. We have now explicitly mentioned this possibility in the revised manuscript. Our numbers are also consistent with the numbers reported in other studies (e.g. Epelboim and Kowler, 1993).

5) Most of the discussion focuses on the neurophysiological mechanisms controlling ocular drift. However, even if short and small, the drift modulation described here may have an impact on, and possibly it may be functional for, visual perception. I would include in the discussion some speculation about the effects of this drift modulation higher up in the visual processing pipeline (e.g., in V1, see for example Snodderly, 2016 on the effects of ocular drift in V1), and about the consequences of this modulation for visual perception.

We have now significantly expanded our Discussion section, as also stated to reviewer 1. This has allowed us to relate our findings to visual processing and perception, and to present a possible functional role for the drift response.

Reviewer #3:This is a technically strong and thorough study of small-scale drift eye movements in macaque monkeys and their relation with visual input. The manuscript is well written, and the analyses are appropriate. The main conclusion is that drift is not random but depends on the visual input. Except for major concern #1 the data support this conclusion.

Thank you very much for this important feedback. We have now addressed major concern #1 below, as we describe in more detail shortly. Briefly, we have now added new sets of experiments with simultaneous eye and head tracking, exactly as suggested. In all experiments, we replicated the eye movement phenomenon without observing a concomitant movement of the head.

Given previously published studies from Michele Rucci's group in collaboration with Jonathan Victor, however, this is not an overly surprising finding. In fact, those publications went substantially further by showing both theoretically and experimentally that drift improves visual function (and not just that visual input affects drift).

As stated above, we have now significantly expanded our Introduction and Discussion sections to clarify the placement of our paper in the literature. We view our results as being highly complementary to other work, including the work of our colleagues Rucci, Victor, and Poletti.

Having said that, we respectfully disagree with your statement that our results are “not overly surprising”. We find our results to be extremely surprising, especially given what they imply for the neurophysiology of the brainstem, as we have alluded to above and in the manuscript (e.g. please see the revised Discussion section). More importantly, the theoretical framework that you have alluded to above (i.e. edge enhancement and spectral whitening by ocular drifts) is actually agnostic of whether these eye movements are random or not; drifts could, in principle, be random walks but still have noise statistics suited for them to functionally aid in edge enhancement and spectral whitening of input retinal images. This is why even some of the Rucci/Victor studies still model drifts as being random walks (e.g. Kuang et al., 2012). Our findings of systematic and directed drift movements (and with such short latencies) are, in our view, very surprising.

Major concern:1) As stated above, the data are strong and the authors have addressed most potential confounds. There is one, however, that appears to be missing. When suddenly exposed to a flash, it could well be that the animals head moves slightly (presumably backward) within the search coil frame. This could cause artefactual position changes due to slight inhomogeneities in the magnetic field. The stimulus dependence that the authors find would then be attributed to the potential to "scare" the animal. For most applications magnetic field homogeneity is good enough, and a head-holder solid enough, but given that we're looking at such small changes in inferred eye position, this remains a possibility that should be addressed. Measurement of head position (using a separate head coil) could address this relatively easily.

We have now conducted new experiments while simultaneously measuring eye and head position. In three different experiments, we placed a head coil at three different locations on the head (to be able to assess multiple possible head motion axes relative to the fixed inducing magnetic fields). Along with calibrations of the head coil positions relative to the monkey chair and head holder (but in the absence of a monkey), we could rule out the possibility that our eye movements were artifactually caused by head movements. Specifically, in all three new experiments, we replicated the eye movement phenomenon that we described in the original manuscript without detecting any concomitant head movements. We are therefore quite confident that our results are not an artifact of head movements. Please see the new Figure 2—figure supplement 1.

Finally, we would like to point out that our flash manipulations were actually quite subtle (e.g. just one single display frame). Such subtlety can be easily confirmed in any psychophysical setup with simple flash paradigms, and it implies that fear or startle are not likely to have caused our results. Rather, we think that there is very high sensitivity of the oculomotor system to certain stimulus onsets. This is a well-known fact in the 20-year-old field of saccadic inhibition (e.g. Figure 4). In that field, we routinely find that a single-frame flip of the luminance of a tiny fixation spot (e.g. from white to black and then back to white in the span of only 8 ms) is sufficient to cause the oculomotor system to react so robustly and so repeatably by completely suppressing microsaccades (Buonocore et al., J. Neurophysiol., 2017). We believe that our results demonstrate a similar sensitivity of ocular position drifts for the types of stimuli that we used.